# Factors Associated with Number of Prenatal Visits in Northeastern Brazil: A Cross-Sectional Study

**DOI:** 10.3390/ijerph192214912

**Published:** 2022-11-12

**Authors:** Gracimary A. Teixeira, Norrara S. O. Holanda, Ingrid G. Azevedo, Julia R. Moura, Jovanka B. L. de Carvalho, Silvana A. Pereira

**Affiliations:** 1Health Technical School, Federal University of Paraiba, João Pessoa 58051-900, PB, Brazil; 2Health Sciences’ Faculty of Trairi, Federal University of Rio Grande do Norte (FACISA—UFRN), Santa Cruz 59200-000, RN, Brazil; 3Department of Therapeutic Processes, Universidad Católica de Temuco, Temuco 4813302, La Araucania, Chile; 4Department of Physical Therapy, Graduate Program of Physical Therapy, Federal University of Rio Grande do Norte (UFRN), Natal 59090-000, RN, Brazil; 5Graduate Program in Health and Society and Graduate Program in Nursing, Federal University of Rio Grande do Norte (UFRN), Natal 59078-970, RN, Brazil

**Keywords:** birth weight, first trimester of pregnancy, prenatal care, preterm newborn

## Abstract

The aim of this study is to assess factors associated with the number of prenatal visits of women who delivered in a public maternity hospital in northeastern Brazil. This cross-sectional study focused on 380 puerperal women who gave birth at a public maternity hospital in northeastern Brazil. Prenatal and perinatal data were collected in the immediate postpartum period by interviewing mothers and using medical records. Chi-square/Fisher exact test compared the data, and a logistic regression model estimated the association between birth weight and number of prenatal visits. As a result, the sample was composed of 175 women with <37 weeks of gestational age and 205 women with ≥37 weeks of gestational age. Women with less than four prenatal visits were more likely to give birth to low birth weight (<2500 g) and preterm infants (<37 weeks of gestational age) than those with more than four prenatal visits (*p* = 0.001). The subjects with less than four prenatal visits had a 2.76-fold higher odds of giving birth to infants weighing less than 2500 g (*p* = 0.03; 95%CI = 1.05–7.30), without relation to maternal and gestational ages. In conclusion, women with less than four prenatal visits had higher odds of giving birth to low birth infants, independently of maternal and gestational ages, and were more likely to give birth to premature babies.

## 1. Introduction

Despite improvements in healthcare, infant morbidity and mortality remain a challenge in Brazil [1,2]. The infant mortality rate is higher in the early neonatal period (i.e., first week after birth), and risk factors include low birth weight, prematurity, low maternal educational level, extremes of maternal age (age < 20), and advanced maternal age (age ≥ 35). Reduction of infant mortality rate requires adequate prenatal, delivery, and postpartum care [1,2].

Premature birth complications were the leading cause of neonatal deaths, and the number of preterm births and low birth weight has increased steadily in recent decades [3,4]. Globally, over 15 million (11.1%) infants are premature [4]. Brazil, United States, India, and Nigeria are among the ten countries with highest preterm delivery rates [3,4,5].

In 2020, prematurity led to 76.21% of early neonatal deaths in Brazil [6]. Northern and northeastern regions of the country recorded the lowest infant mortality rates (69.31% and 74.54%, respectively), below national average, while southeastern (79.49%), central-western (78.05%), and southern (77.11%) were above national average [6]. In the same year, in Brazil, preterm accounted for 12.46% of births for 12.46% of births [6].

Preterm birth is the result of preconception or gestational issues and interrelated factors, such as social aspects, race, low quality and restricted access to prenatal care, excessive medical interventions during vaginal delivery due to error diagnosing onset of labor, elective cesarean delivery, and inadequate determination of gestational age [7,8,9,10,11,12,13].

Healthcare during pregnancy-puerperal cycle allows monitoring risk factors for infant mortality rate and is critical for improving perinatal outcomes [14,15]. Health professionals must provide adequate prenatal care (more than four visits), including welcoming environment, establishment of educational strategies, early detection of pathologies and high-risk pregnancy [14,16,17,18,19].

Studies performed in South Africa found that most puerperal women do not receive prenatal care during the first trimester, and its onset is associated with maternal education and parity [17,18,19,20]. These correlations were also found in southeastern Brazil [19]. Given that northeastern Brazil has one of the lowest literacy rates and highest adolescent pregnancy rates in the country [11], this study aimed to evaluate factors related to the number of prenatal visits of women who delivered in a public maternity hospital in northeastern Brazil.

## 2. Materials and Methods

This cross-sectional study assessed 380 puerperal women who gave birth in a public maternity hospital in northeastern Brazil (Hospital Maternidade Divino Amor) between April 2018 and March 2019. The maternity hospital is located in the city of Parnamirim, in the state of Rio Grande do Norte. This municipality has established the “Stork Network” (Rede Cegonha), a national program that offers 100% prenatal care coverage by the Family Health Strategy for pregnant women with a high-risk pregnancy. The Stork Network is a strategy that aims to structure and organize maternal and child healthcare in Brazil and it is composed of four components: (I)-Prenatal; (II)-Childbirth and birth; (III)-Puerperium and integral child healthcare; and (IV)-Logistics system (sanitary transport and regulation).

The study has been approved by the research ethics committee of the Federal University of Rio Grande do Norte (UFRN) (no. 1.047.431/2015) and performed in compliance with the Declaration of Helsinki. All participants signed the informed consent form. Pregnant adolescents were included if they provided assent and had signed consent by legal parent or guardian.

Both puerperal women residents in the city of the public maternity hospital and who gave birth there were included. Puerperal women with no record of gestational age or with possible mental disorders were excluded as they could not consent for themselves. Prenatal and perinatal data were collected in the immediate postpartum period (first 24 h postpartum) by interviewing mothers and using medical records.

Birth weight, maternal and gestational age were collected from medical records, while information regarding the income and education were collected by interviewing the mother.

The primary outcome was the number of prenatal visits and the study variables were distributed into categories as follows:

Number of prenatal visits: Less than 4 or equal/higher than 4 visits [17,21].

Maternal age: This variable was divided into 3: ≤19 years, between 20–34 years, and ≥35 years [2,17].

Birth weight: This variable was categorized into low birth weight (<2500 g) and birth weight ≥2500 g [21].

Gestational age: Preterm infants were considered those born alive before 37 completed weeks of gestation (<37 weeks of gestational age), while full-term were considered the infants born with ≥37 weeks [4].

Maternal education: The women’s schooling was categorized into < than elementary school or ≥elementary school [4].

Family income: This variable was categorized according to minimum monthly wage: ≥1; 1–2, or 3–5 minimum monthly wages [17].

### Statistical Analyses

For data analysis, frequencies distribution and percentages were used for the categorical variables. Chi-square test or Fisher exact test (for the categories with less than 5 cases) compared prenatal/perinatal data and maternal variables. A *p* < 0.05 was considered as statistically significant. A logistic regression model was adjusted to estimate associations between number of prenatal visits (<4 visits) and variables that presented *p* < 0.20 in bivariate analysis, i.e., maternal age, birth weight, and gestational age. Since none of the variables considered in the study were associated with the time of the first prenatal visit, logistic regression was performed only for the variable ‘number of prenatal visits’. Statistical analysis was undertaken using SPSS 20 (SPSS, Inc., Chicago, IL, USA).

## 3. Results

Three hundred and eighty puerperal women were evaluated: 175 puerperal women with <37 weeks of gestational age and 205 puerperal women with ≥37 weeks of gestational age. Most women (93%) received public prenatal care, and almost 7% received care from the private sector. Only four of the included women (0.019%) did not receive prenatal care.

Table 1 shows the number of visits and the trimester of first prenatal visit according to delivery-related variables (birth weight, gestational age at birth, maternal age, family income, and maternal education). Women with less than four prenatal visits were more likely to give birth to low birth weight (<2500 g) and preterm infants (<37 weeks of gestational age) than those with more than four prenatal visits (*p* = 0.001). The trimester of the first prenatal visit (first, second, or third trimester) was not associated with the delivery conditions (gestational age at birth and birth weight).

Adherence to prenatal care in the first trimester of pregnancy was higher in women aged between 20 and 34 years (60.6) when compared to the other women groups (≤19 years and ≥35 years) However, these results were not statistically significant (*p* = 0.08). Additionally, 72.8% of those who had less than four prenatal visits (*n* = 33) aged between 20–34 years (*n* = 24) Preterm birth was more frequent (72.7%) in women with less than four prenatal visits (*p* < 0.001).

Table 2 shows that women with less than four prenatal visits had a 2.76-fold higher odds of giving birth to infants weighing less than 2500 g (*p* = 0.03; 95%CI = 1.05–7.30), independently of maternal and gestational ages.

## 4. Discussion

In the studied sample, having less than four prenatal visits is associated with almost 3-fold higher odds of low birth weight. Almost 73% of those who had less than four prenatal visits were aged between 20–34 years. Additionally, preterm birth was more frequent (72.7%) in women with less than four prenatal visits. Prenatal care should ensure a healthy pregnancy for both the mother and infant. Educational and preventive strategies addressing psychosocial aspects of prenatal period, delivery, and postpartum should be introduced by health professionals in the first visits, regardless of trimester of pregnancy [21].

Furthermore, 66% of women completed elementary school or higher education, and 68% had income between one and two minimum monthly wages (~USD 250–500). A study of Angolan women found that inadequate prenatal care was more likely among young women with low levels of education and income [20].

Insufficient visit frequency increased almost 3-fold the odds of low birth weight, an important indicator of infant health status [22]. Low birth weight has been associated with several factors, such as mothers aged <20 or >35 years, maternal malnutrition, genitourinary tract infection during pregnancy, previous low birth weight babies, premature delivery, birth interval less than 18 months, smoking during pregnancy, cesarean delivery, and low maternal education [11,13,23,24].

However, no study has linked the number of prenatal consultations to the risk of low birth weight, emphasizing the importance and innovation of our study for maternal and infant care. Our results show that low birth weight was associated with the number of prenatal visits regardless of maternal and gestational ages. This backs the importance of following the recommendations of the number of prenatal visits, since the infant mortality rate is higher in low birth weight [1,24]. Because neonatal deaths in Brazil have not decreased as expected, mainly due to perinatal infections [21], improvements in prenatal, delivery, and perinatal care are needed.

Taking into account the well-qualified healthcare needed to decrease the number of deaths among high-risk factor groups, maternal age is an important variable that should be controlled, since adolescent childbirth is related to higher odds of low birth weight, and, consequently, to mortality [16]. In our sample, young adult mothers (20–34 years) were less likely to have low birth weight babies when compared to adolescent mothers, corroborating other literatures [2,11,16,17] This age group (20–34 years) is associated with less risks among other variables related to neonatal outcomes in other studies [2,16,17].

This study presents some strengths: to identify the factors that may be related to low birth weight and premature birth, which are causes of mortality, and then address the necessary healthcare. Regrettably, we did not address associations between content and quality of prenatal visits and birth outcomes, highlighting the limitations for this study. Limited access to prenatal care, including issues related to medical records, may hinder favorable birth outcomes. Inadequate educational and prevention strategies during prenatal care compromise pregnant women’s health and child development [19]. In northeastern Brazil, improvements in prenatal and perinatal services are urgently needed to ensure adequate and comprehensive care for pregnant women. Identification of pregnant women, especially those at extreme ages, is essential to provide early prenatal care and reduce the infant mortality rate. Further studies may investigate associations between quality of antenatal care and childbirth.

## 5. Conclusions

In the studied sample from northeastern Brazil, women with fewer than four prenatal visits had higher odds of giving birth to low birth babies (<2500 g) when compared to women who had four or more visits, regardless of maternal and gestational ages. Furthermore, women with fewer than four prenatal visits were more likely to give birth to premature infants (<37 weeks of gestational age) than those with more than four prenatal visits.

## Figures and Tables

**Table 1 ijerph-19-14912-t001:** Sample characterization according to prenatal visits.

	Number of Visits–*n* (%)			First Prenatal Visit *–*n* (%)		
	Four or More	Less than Four	*p*-Value	1st Trimester	2nd/3rd Trimester	*p*-Value
Birth Weight *			0.001			
<2500 g	82 (24.0%)	18 (56.3%)		65 (38.7%)	10 (32.3%)	0.32
≥2500 g	259 (76.0%)	14 (43.8%)		103 (61.3%)	21 (67.7%)	
Maternal Age			0.06			
≤19 years	87 (25.1%)	3 (9.0%)		43 (24.9%)	8 (25%)	0.51
Between 20 and 34 years	205 (59.1%)	24 (72.8%)		105 (60.6%)	20 (62.5%)	
≥35 years	55 (15.9%)	6 (18.2%)		25 (14.5%)	4 (12.5%)	
Maternal Education			0.56			
<elementary school	119 (34.3%)	10 (30.3%)		62 (35.8%)	13 (40.6%)	0.37
≥elementary school	228 (65.7%)	23 (69.7%)		111 (64.2%)	19 (59.4%)	
Family Income			0.68			
≥1 minimum monthly wage	33 (9.5%)	1 (3.0%)		21 (12.1%)	3 (9.4%)	0.64
Between 1 and 2 minimum monthly wages	232 (66.9%)	26 (78.8%)		113 (65.4%)	24 (75%)	
Between 3 and 5 minimum monthly wages	82 (23.6%)	6 (18.2%)		39 (22.5%)	5 (15.6%)	
Gestational age			0.001			
Preterm	151 (43.5%)	24 (72.7%)		99 (57.2%)	16 (50%)	0.295
Full-term	196 (56.5%)	9 (27.3%)		74(42.8%)	16 (50%)	

* Missing data for these variables. For the first prenatal visit, the total sample was 205 subjects. However, only 199 of them had data regarding birth weight.

**Table 2 ijerph-19-14912-t002:** Logistic regression for variables associated with number of prenatal visits (<4 visits).

Variables	Standard Error	OR	*p*-Value
Maternal Age			
0–19 years	1	1	-
20–34 years	0.830	5.36	0.043
35 years or more	0.474	1.27	0.614
Gestational Age (<37 weeks)	0.533	0.61	0.367
Birth Weight (<2500 g)	0.494	2.76	0.039
Constant	0.663	5.22	0.013
Equation Model	0.188	11.000	*p* < 0.0001

OR: odds ratio.

## Data Availability

Not applicable.

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
