# Peer review of "Factors Associated with Number of Prenatal Visits in Northeastern Brazil: A Cross-Sectional Study"

_ijerph, 2022, doi:10.3390/ijerph192214912_

Round 1
Reviewer 1 Report
Thank you for the invitation for reviewing this paper. I have some comments and suggestions for the authors as described below.
Overall comments:
This is a cross-sectional study, so there is impossible to analyze causality. Thus, I suggest the authors to change all words related to causality from the paper. For instance, the authors say that the objective is to analyze the effect of prenatal care on infant’s weight. However, the authors can only say that there is an association between prenatal care and birth weight. To determine the effect, the authors would need a randomized controlled trial. Also, the results must describe the associations found, which represent higher odds of the outcome, not higher risk.
Also, after reading the entire paper, I suggest the authors change the title and objective to match with the statistical analysis and results presented, which is to assess factors associated with number of prenatal visits. The association between low birth weight and prenatal visits was only one of the factors analyzed, so it should not be described as the main objective. If the authors want to keep this as main objective, they should do the regression models including birth weight as outcome.
Abstract:
Besides the comments above, the abstract lacks information about the variables considered in the analyses. The result about prenatal adherence in the first trimester makes no sense as it is. Why describe this result for only one age group? Are there other age groups? It is also not clear the conclusion when the authors say about young pregnant women. Which age group they are referring to? What are the results for the other age groups?
Introduction:
The introduction is clear and describes well the rationale under the study’s objective. I have only few suggestions. Please include references for the first two sentences of the second paragraph. They bring statistics, so it is important to know from where they were retrieved. Same thing with the first sentence of the third paragraph. The authors should also include a reference for the information on health literacy rates and adolescent pregnancy rates in Northeastern Brazil in the last sentence of the introduction.
Methods:
Please describe a little bit more about the Stork Network. Is this a national or local program? Only high-risk pregnant women are covered by this program?
The authors describe that receiving prenatal care is one of the inclusion criteria. Is this correct? If the objective was to analyze the association between prenatal care and birth weight, why exclude those with no prenatal care? In the results, the authors describe that only 4 did not receive prenatal care. But it is not clear if they were excluded of this study or not.
Adolescent women were also included? If yes, did the authors request consent from their legal representatives? Did they sign an assent form? Please, provide more details.
The methods should include the variables collected, and which variables were collected by interviewing the mother and which were collected from the medical records. They should also describe the groups created for the variables under study, so the reader can understand better the results. For instance, why the authors chose to categorize prenatal visits as less or more than 4? Also, those who had exactly 4 visits, belong to which group? It is also important to understand which variables were considered as potential covariates of the study.
Regarding birth weight, how was it considered in the analysis? Please, describe the categories created in the methods. Please, provide more details in the statistics. There is an information of p<0.05 after the first sentence. This means that the analysis considered p<0.05 as significant in the analyses? Please, describe this properly. If the authors want to keep as main objective the evaluation of the association between prenatal visits and low birth weight, the variable regarding low birth weight should be the outcome in the regression analysis, not the prenatal visits. So, it would be better to change the title and objective as I described above. Additionally, the authors performed logistic regression only for the number of prenatal visits. Why did they not present the same analysis for the prenatal visit in the first trimester? If it was because none of the variables considered in the study were associated with time of the first prenatal visit, the authors should let this clearer in the statistics topic.
Results:
Please, change the word “influence” in the last sentence of the first paragraph. It is not possible to determine influence in a cross-sectional study. Use terms related to “association” throughout the text.
Please correct the information regarding maternal education in the table and regarding the age groups (first category should be up to 19, not more than 19).
The authors said they evaluated 308 women in the results, but the number of participants in the tables are of around 380. Please, correct the text. Also, in table 1, there are 373 infants described in the variable related to birth weight and 381 in the variable related to maternal age. Please, check all numbers in the tables and add footnotes to describe any missing data.
Moreover, there are some results with less than 5 participants in some categories, so the authors should use the Fisher exact test for those cases. Please, correct this.
This sentence “adherence to prenatal care in the first trimester was higher in pregnant women aged between 20 and 34 years (62.5%)” does not match with the results in the table. The result was not significant (p=0.08) and the percentage of women with prenatal care in the first trimester with age between 20 and 34 years was 60%. Moreover, the sum of the percentages for age group are not reaching 100%, so there is something wrong with the number or there is some missing data not reported.
Also, the sentence “However, 70% of those had less than four prenatal visits” is not accurate. The table shows that 70.6% of the participants with less than four visits are from the group between 20 and 34 years of age, not that 70% of participants if this age group had less than 4 prenatal visits. If you do the math, there is 229 participants in this group and only 24 had less than 4 prenatal visits, with represents a little bit more than 10% of those women.
Describe the results in table 2 as higher odds, not higher risk, i.e., women with less than four prenatal visits had a 2.76-fold higher odds of having given birth to infants weighing less than 2500 g.
In table 2, change the zeros for the reference category of maternal age for 1. Also, let clear which group of gestational age and birth weight are being presented in the table.
Discussion
Correct the first sentence, since it is not accurate as already described in the section above. Change the words related to risk as well.
In this sentence “Furthermore, 70% of women completed elementary school or higher education, and 79% had income between one and two minimum monthly wages (~USD 250-500)” who are the authors referring to? If they are considering the total sample, this number is equal 66% (251 of 380).
The discussion should be more informative in relation to the main findings. One thing the authors should comment in the discussion if that low birth weight was associated with number of prenatal visits irrespectively of maternal age and gestational age, which are variables that could confound the results. They should also include discussion on the results regarding maternal age. Young-adult mothers (20-34 years) presents lower odds of infant with low birth weight compared to adolescent mothers. This age group is associated with less risks among other variables related to neonatal outcomes in other studies. Also, this mean that adolescent mothers are in the group with higher odds of low birth weight, and there are no comments about this in the text. Adolescent childbirth is associated with low birth weight irrespective of number of prenatal visits. The authors should call the attention of the need for high quality care among this group.
Author Response
Dear Reviewer,
The manuscript entitled “Factors associated with number of prenatal visits in Northeastern Brazil: a cross-sectional study” has been reviewed according to the reviewers comments and it is being re-submitted to be considered for publication in the International Journal of Environmental Research and Public Health. Considering all the points related by the reviewers, we have reviewed the paper and alterations have been made as it follows:
REVIEWER 1
Overall comments:
This is a cross-sectional study, so there is impossible to analyze causality. Thus, I suggest the authors to change all words related to causality from the paper. For instance, the authors say that the objective is to analyze the effect of prenatal care on infant’s weight. However, the authors can only say that there is an association between prenatal care and birth weight. To determine the effect, the authors would need a randomized controlled trial. Also, the results must describe the associations found, which represent higher odds of the outcome, not higher risk.
Response: Thank you for the comment. We have accepted the suggestions and corrected the terms.
Also, after reading the entire paper, I suggest the authors change the title and objective to match with the statistical analysis and results presented, which is to assess factors associated with number of prenatal visits. The association between low birth weight and prenatal visits was only one of the factors analyzed, so it should not be described as the main objective. If the authors want to keep this as main objective, they should do the regression models including birth weight as outcome.
Response: Thank you for the comment. We have accepted the suggestions.
Abstract:
Besides the comments above, the abstract lacks information about the variables considered in the analyses. The result about prenatal adherence in the first trimester makes no sense as it is. Why describe this result for only one age group? Are there other age groups? It is also not clear the conclusion when the authors say about young pregnant women. Which age group they are referring to? What are the results for the other age groups?
Answer: Thanks for the comments. We have rewritten the abstract.
Introduction:
The introduction is clear and describes well the rationale under the study’s objective.
Response: Thank you very much. We appreciate .
I have only few suggestions. Please include references for the first two sentences of the second paragraph. They bring statistics, so it is important to know from where they were retrieved. Same thing with the first sentence of the third paragraph. The authors should also include a reference for the information on health literacy rates and adolescent pregnancy rates in Northeastern Brazil in the last sentence of the introduction.
Response: Thank you for the comment. We added all the required references.
Methods:
Please describe a little bit more about the Stork Network. Is this a national or local program? Only high-risk pregnant women are covered by this program?
Answer: The Stork Network is a national program that aims to structure and organize maternal and child health care in Brazil and it is composed for four components: I - Prenatal; II - Childbirth and birth; III - Puerperium and integral child healthcare; and IV - Logistics system (sanitary transport and regulation). It offers 100% prenatal care coverage by the Family Health Strategy for pregnant women at high-risk pregnancy (Information added to LINES 85-89).
The authors describe that receiving prenatal care is one of the inclusion criteria. Is this correct? If the objective was to analyze the association between prenatal care and birth weight, why exclude those with no prenatal care? In the results, the authors describe that only 4 did not receive prenatal care. But it is not clear if they were excluded of this study or not.
Response: We apologize. Indeed, receiving prenatal care was not an inclusion criteria.
The 4 subjects who did not receive prenatal care were included in the study. We have corrected the criteria (LINE 94-95 + LINE 135).
Adolescent women were also included? If yes, did the authors request consent from their legal representatives? Did they sign an assent form? Please, provide more details.
Response: The adolescent subjects had their consent form signed by their legal representatives.
The methods should include the variables collected, and which variables were collected by interviewing the mother and which were collected from the medical records. They should also describe the groups created for the variables under study, so the reader can understand better the results. For instance, why the authors chose to categorize prenatal visits as less or more than 4? Also, those who had exactly 4 visits, belong to which group? It is also important to understand which variables were considered as potential covariates of the study.
Response: We are sorry. In fact, we considered the categories of < 4 or ≥ 4 visits, as it is in the table 1. Plus, we added this into the methods. We have corrected the information in the statistical analysis: for the regression, the category of < 4 visits was considered. Then, we kept the number of visits < 4 for the logistic regression (LINES 125-129). Regarding the variable categories, we have described all of them in the methods.
Regarding birth weight, how was it considered in the analysis? Please, describe the categories created in the methods. Please, provide more details in the statistics. There is an information of p<0.05 after the first sentence. This means that the analysis considered p<0.05 as significant in the analyses? Please, describe this properly.
Response: We appreciate your suggestion. We described the categories properly in the methods (LINES 100-117 + LINES 121-130).
If the authors want to keep as main objective the evaluation of the association between prenatal visits and low birth weight, the variable regarding low birth weight should be the outcome in the regression analysis, not the prenatal visits. So, it would be better to change the title and objective as I described above.
Response: We appreciate your suggestion. We changed the title and the main objective of the study.
Additionally, the authors performed logistic regression only for the number of prenatal visits. Why did they not present the same analysis for the prenatal visit in the first trimester? If it was because none of the variables considered in the study were associated with time of the first prenatal visit, the authors should let this clearer in the statistics topic.
Response: Thanks for your suggestion. We clarified it in the statistics topic (LINES 121-130).
Results:
Please, change the word “influence” in the last sentence of the first paragraph. It is not possible to determine influence in a cross-sectional study. Use terms related to “association” throughout the text.
Response: Thank you for the comment. We accepted the suggestion.
Please correct the information regarding maternal education in the table and regarding the age groups (first category should be up to 19, not more than 19).
Response: We have corrected both.
The authors said they evaluated 308 women in the results, but the number of participants in the tables are of around 380. Please, correct the text.
Response: We have corrected.
Also, in table 1, there are 373 infants described in the variable related to birth weight and 381 in the variable related to maternal age. Please, check all numbers in the tables and add footnotes to describe any missing data.
Response: The variable related to birth weight has 7 missing cases. We added as footnotes. Pl us, we corrected the number 381 to 380, regarding the variable maternal age.
Moreover, there are some results with less than 5 participants in some categories, so the authors should use the Fisher exact test for those cases. Please, correct this.
Response: Thanks for the consideration. We have corrected that.
This sentence “adherence to prenatal care in the first trimester was higher in pregnant women aged between 20 and 34 years (62.5%)” does not match with the results in the table. The result was not significant (p=0.08) and the percentage of women with prenatal care in the first trimester with age between 20 and 34 years was 60%. Moreover, the sum of the percentages for age group are not reaching 100%, so there is something wrong with the number or there is some missing data not reported.
Response: Thanks for the comment. We have corrected the frequency and percentage of the age group; we also have added the missing data to the variables “birth weight” and “first prenatal visit”.
Also, the sentence “However, 70% of those had less than four prenatal visits” is not accurate. The table shows that 70.6% of the participants with less than four visits are from the group between 20 and 34 years of age, not that 70% of participants if this age group had less than 4 prenatal visits. If you do the math, there is 229 participants in this group and only 24 had less than 4 prenatal visits, with represents a little bit more than 10% of those women.
Response: We calculated 24 subjects from a total of 33 subjects (72.7%), related to the column that considered the mothers with < 4 prenatal visits (n=33).
Describe the results in table 2 as higher odds, not higher risk, i.e., women with less than four prenatal visits had a 2.76-fold higher odds of having given birth to infants weighing less than 2500 g.
Response: We have corrected (LINES 152-154).
In table 2, change the zeros for the reference category of maternal age for 1. Also, let clear which group of gestational age and birth weight are being presented in the table.
Response: We have added the required information to the table 2.
Discussion
Correct the first sentence, since it is not accurate as already described in the section above.
Answer: Thank you. We have corrected.
Change the words related to risk as well.
Answer: We have changed.
In this sentence “Furthermore, 70% of women completed elementary school or higher education, and 79% had income between one and two minimum monthly wages (~USD 250-500)” who are the authors referring to? If they are considering the total sample, this number is equal 66% (251 of 380).
Answer: Thank you. We have corrected.
The discussion should be more informative in relation to the main findings. One thing the authors should comment in the discussion if that low birth weight was associated with number of prenatal visits irrespectively of maternal age and gestational age, which are variables that could confound the results.
Answer: Thanks for the suggestion. We have added it.
They should also include discussion on the results regarding maternal age. Young-adult mothers (20-34 years) presents lower odds of infant with low birth weight compared to adolescent mothers. This age group is associated with less risks among other variables related to neonatal outcomes in other studies. Also, this mean that adolescent mothers are in the group with higher odds of low birth weight, and there are no comments about this in the text. Adolescent childbirth is associated with low birth weight irrespective of number of prenatal visits. The authors should call the attention of the need for high quality care among this group.
Answer: Thanks for the suggestion. We have added the information.
Reviewer 2 Report
Using a cross-sectional design, the authors aimed to evaluate the role of prenatal care practices on birth outcomes in Northern Brazil.
The manuscript would benefit from thorough proof-reading for language and grammar, though overall it reads well.
Additionally, the manuscript would be stronger with the addition of more birth data. I have provided several specific comments below.
Line 29-30: This sentence does not read clearly and should be revised.
Line 48: Please include the source for this data.
Line 52: "...accounted for 12.46%." to what is the 12.46% referring? Is this saying prematurity accounted for 12.46% of infant mortality? Please clarify accordingly.
Line 71: I would include the name of the hospital.
Line 77: Include the location, ie the city that the study was performed in if possible.
Line 79: Women with possible mental illness were excluded. I assume this was because they could not consent for themselves. Perhaps clarify the reason for this exclusion in the text.
Statistical Analysis
Line 84: Were all maternal variables categorical? Chi-square test should not be used for continuous data.
Lines 83-87: What was the primary outcome and what are the secondary outcomes selected for the analysis? This is not clearly defined in the text.
Line 87: Why was birth weight not evaluated as continuous variables in addition to the existing analysis? I think that this might offer additional power and insight into the differences in birth outcomes between the groups.
Additionally, are other birth anthropometric measures (length) available for analysis? These other measures can be important indicators for morbidity and mortality for neonates. The manuscript feels a bit incomplete with only the LBW analysis and no other birth anthro, including actual birth weight data. Use of continuous variables would add additional power to the study.
Line 89: During what time period were the women enrolled? This information should be included in the text.
Table 2: Perhaps replacing the "0" in the. "up to 19 years" fields with the text "Reference" might make more sense to readers.
Discussion
Line 109: The discussion is quite brief in its current form. I would suggest adding more Discussion especially around the context in Brazil and how this fits with other studies in similar upper middle income settings.
Additionally, the authors. could explore some of the confounding factors that contribute to preterm birth that may not have been evaluated in the current study.
Line 133: The authors note a limitation in this paragraph. I would suggest they describe strengths and limitations of the study they performed to make this easy for the reader.
Conclusion
Line 145: I would suggest adding the context of the study in this line as it current reads broadly generalized and may not actually be the case in other settings.
Author Response
Dear Editors-Reviewer,
The manuscript entitled “Factors associated with number of prenatal visits in Northeastern Brazil: a cross-sectional study” has been reviewed according to the reviewers comments and it is being re-submitted to be considered for publication in the International Journal of Environmental Research and Public Health. Considering all the points related by the reviewers, we have reviewed the paper and alterations have been made as it follows:
REVIEWER 2
The manuscript would benefit from thorough proof-reading for language and grammar, though overall it reads well.
Response: Thank you very much. The manuscript has been reviewed.
Additionally, the manuscript would be stronger with the addition of more birth data. I have provided several specific comments below.
Line 29-30: This sentence does not read clearly and should be revised.
Response: We rewrote the sentence.
Line 48: Please include the source for this data.
Response: Thank you for the comment. We have added all the required references.
Line 52: "...accounted for 12.46%." to what is the 12.46% referring? Is this saying prematurity accounted for 12.46% of infant mortality? Please clarify accordingly.
Answer: We have clarified the sentence.
In the same year (2021), in Brazil, preterm accounted for 12.46% of births.
Line 71: I would include the name of the hospital.
Response: We have added all the required information.
Hospital Maternidade Divino Amor (LINE 81).
Line 77: Include the location, ie the city that the study was performed in if possible.
Response: We have added all the required information:
This cross-sectional study assessed 380 puerperal women who gave birth in a public maternity hospital in Northeastern Brazil (Hospital Maternidade Divino Amor) between April 2018 and March 2019. The maternity hospital is located in the city of Parnamirim, in the state of Rio Grande do Norte.
Line 79: Women with possible mental illness were excluded. I assume this was because they could not consent for themselves. Perhaps clarify the reason for this exclusion in the text.
Response: We have added all the required information (LINE 97).
Statistical Analysis
Line 84: Were all maternal variables categorical? Chi-square test should not be used for continuous data.
Answer: All the variables were categorical. We did not use Chi-square for continuous data.
Lines 83-87: What was the primary outcome and what are the secondary outcomes selected for the analysis? This is not clearly defined in the text.
Response: The primary outcome was the number of prenatal visits. We have clarified this in the text (LINES 100-116).
Line 87: Why was birth weight not evaluated as continuous variables in addition to the existing analysis? I think that this might offer additional power and insight into the differences in birth outcomes between the groups.
Response: We appreciate the comment.
We splitted the sample between < 2500 and ≥ 2500 grams, i.e., in two categories. For the next researches we will considered to evaluate as a continuous variable.
Additionally, are other birth anthropometric measures (length) available for analysis? These other measures can be important indicators for morbidity and mortality for neonates. The manuscript feels a bit incomplete with only the LBW analysis and no other birth anthro, including actual birth weight data. Use of continuous variables would add additional power to the study.
Response: We appreciate the comment and we will consider them for next researches.
Line 89: During what time period were the women enrolled? This information should be included in the text.
Response: The women had been enrolled within their immediate puerperium (the first 24 hours after birth). We added this information (LINE 98).
Table 2: Perhaps replacing the "0" in the. "up to 19 years" fields with the text "Reference" might make more sense to readers.
Answer: Thank you. We replaced.
Discussion
Line 109: The discussion is quite brief in its current form. I would suggest adding more Discussion especially around the context in Brazil and how this fits with other studies in similar upper middle income settings.
Additionally, the authors. could explore some of the confounding factors that contribute to preterm birth that may not have been evaluated in the current study.
Answer: Thanks for the suggestion. We have improved the discussion.
Line 133: The authors note a limitation in this paragraph. I would suggest they describe strengths and limitations of the study they performed to make this easy for the reader.
Response: We have added the required information.
Conclusion
Line 145: I would suggest adding the context of the study in this line as it current reads broadly generalized and may not actually be the case in other settings.
Response: We have added the required information.
Round 2
Reviewer 1 Report
Thank you for your hard work on this paper. I have some two comments about the current version.
Lines 135 to 137 do not match with the results in the tables.
Line 146 – change “increased almost 3-fold”for “it is associated with almost 3-fold higher odds”.
Author Response
Dear Editors-Reviewer,
The manuscript entitled “Factors associated with number of prenatal visits in Northeastern Brazil: a cross-sectional study” has been reviewed according to the reviewers comments and it is being re-submitted to be considered for publication in the International Journal of Environmental Research and Public Health. Considering all the points related by the reviewers, we have reviewed the paper and alterations have been made as it follows:
REVIEWER 1
Thank you for your hard work on this paper. I have some two comments about the current version.
Lines 135 to 137 do not match with the results in the tables.
Response: Thank you very much. The number of women with <4 prenatal visits was 33.
72.8% of 33 = 24 women. We have changed the sentence to try to make it clearly: Additionally, 72.8% of those who had less than four prenatal visits (n=33) aged between 20-34 years (n=24).
Line 146 – change “increased almost 3-fold”for “it is associated with almost 3-fold higher odds
Response: Thank you very much. We have changed it (line 165).
Reviewer 2 Report
Thank you for your revision of the manuscript. The majority of my concerns and comments have been addressed. The manuscript would benefit from thorough editing for English grammar as multiple passages have awkward flow and unusual sentence structure.
I have provided a few specific comments below:
Line 53: This sentence doesn't make sense. It appears to repeat the same information twice.
Line 83: This sentence currently reads weird. I would suggest editing to say "Pregnant adolescents were included if they provided assent and had signed consent by legal parent or guardian."
Lines 105-107: I would suggest moving this sentence up to Line 90.
Line 129: I would suggest replacing "semester" with "trimester".
Line 146-147: This line reads awkwardly.
Table 1: For the footnote, I would suggest placing the note all in the same line as all the text speaks to is missing data and correlates with the * symbol placed in the table header.
Table 2 Title needs more descriptive text so the reader can interpret without referring to the main text, so the number of visit classification should be provided.
Author Response
Dear Editors-Reviewer,
The manuscript entitled “Factors associated with number of prenatal visits in Northeastern Brazil: a cross-sectional study” has been reviewed according to the reviewers comments and it is being re-submitted to be considered for publication in the International Journal of Environmental Research and Public Health. Considering all the points related by the reviewers, we have reviewed the paper and alterations have been made as it follows:
REVIEWER 2
Thank you for your revision of the manuscript. The majority of my concerns and comments have been addressed. The manuscript would benefit from thorough editing for English grammar as multiple passages have awkward flow and unusual sentence structure.
Answer: Thank you. For the second time, we have asked for an English review, made by a translation company.
I have provided a few specific comments below:
Line 53: This sentence doesn't make sense. It appears to repeat the same information twice.
Answer: Thanks. We have erased it.
Line 83: This sentence currently reads weird. I would suggest editing to say "Pregnant adolescents were included if they provided assent and had signed consent by legal parent or guardian."
Answer: We have replaced it.
Lines 105-107: I would suggest moving this sentence up to Line 90.
Answer: We moved it. (in the new lines numbers, they are on line 101, not 90).
Line 129: I would suggest replacing "semester" with "trimester".
Answer: We have replaced (line 143)
Line 146-147: This line reads awkwardly.
Answer: We have improved.
Table 1: For the footnote, I would suggest placing the note all in the same line as all the text speaks to is missing data and correlates with the * symbol placed in the table header.
Answer: We have placed it all in the same line.
Table 2 Title needs more descriptive text so the reader can interpret without referring to the main text, so the number of visit classification should be provided.
Answer: We have added the number of visits.